# Structural Stabilization of Mullite Films Exposed to Oxygen Potential Gradients at High Temperatures

**Satoshi Kitaoka \*, Tsuneaki Matsudaira, Naoki Kawashima, Daisaku Yokoe, Takeharu Kato and Masasuke Takata**

Japan Fine Ceramics Center (JFCC), Nagoya 456-8587, Japan; matsudaira@jfcc.or.jp (T.M.); kawashima@jfcc.or.jp (N.K.); yokoe@jfcc.or.jp (D.Y.); tkato@jfcc.or.jp (T.K.); m_takata@jfcc.or.jp (M.T.)
* Correspondence: kitaoka@jfcc.or.jp; Tel.: +81-52-871-3500

**Abstract:** The oxygen shielding properties of polycrystalline $Al_{4+2x}Si_{2-2x}O_{10-x}$ (mullite) films applied as environmental barrier coatings (EBCs) on SiC fiber-reinforced SiC matrix composites (SiC/SiC) are determined by the grain boundary (GB) diffusion of oxide ions in the films, from the higher oxygen partial pressure ($P_{O_2}$) surface to the lower $P_{O_2}$ surface, with simultaneous GB diffusion of Al ions in the opposite direction. Herein, strategies to improve the oxygen shielding and phase stability of these films when applied to SiC/SiC substrates through bond coats are proposed, based on oxygen permeation data for mullite at high temperatures. The validity of these strategies is verified using experimental trials at 1673 K with bilayer specimens consisting of mullite films and bond coat substrates, serving as model EBCs. The data show that employing a bond coat made of β'-SiAlON rather than Si provides a source of Al for the overlying mullite film that greatly improves the phase stability of the film in the vicinity of the junction interface. Because the minimum equilibrium $P_{O_2}$ values required to form $SiO_2$ due to oxidation of the β'-SiAlON on a thermodynamic basis are significantly larger than those for oxidation of Si, the inward GB diffusion of oxide ions is effectively retarded, resulting in excellent oxygen shielding characteristics.

**Keywords:** EBCs; mullite; SiAlON; diffusion; grain boundary; oxygen permeability

## 1. Introduction

Environmental barrier coatings (EBCs) can play a key role in enabling SiC fiber-reinforced SiC matrix (SiC/SiC) composites to be employed as advanced hot-section components in airplane engines, as a means of realizing exceptional fuel efficiency. EBCs must exhibit excellent oxygen/water vapor shielding characteristics and thermomechanical durability in severe combustion environments [1–7]. Thus, a multilayer structure is generally adopted, with each layer having its own unique properties. Naturally, EBCs are exposed to a large oxygen potential gradient ($d\mu_O$) at elevated temperatures. This gradient results in the inward diffusion of oxide ions and outward diffusion of cations, according to the Gibbs-Duhem relationship. Cation transport through an oxide layer typically induces decomposition of the oxide, and can thus cause the EBC structure to collapse. Therefore, the development of a robust EBC with excellent gas shielding characteristics requires both a good understanding of, and control over, mass transfer within the EBC. However, even though the performance of an EBC is greatly affected by both its constituent materials and its microstructure depending on process conditions, there is a crucial lack of knowledge regarding the environmental shielding properties of these materials. Consequently, studies intended to improve EBC performance tend to be based on trial-and-error.

In prior work, our group elucidated the mass transfer processes within the constituent oxides of EBCs by determining the oxygen permeability values of wafers cut from sintered bodies serving as model EBCs [8–13]. This technique is believed to accurately evaluate the environmental shielding

characteristics of EBC materials under steady state conditions. This is possible because both the $d\mu_O$ applied to the wafers and the diffusion length are kept constant. In this technique, the upper and lower surfaces of the wafer are exposed to gases with different oxygen partial pressures ($P_{O_2}$) such that the dissociative adsorption of oxygen molecules will proceed at the surface having the higher $P_{O_2}$ (the $P_{O_2}$(hi) surface), while the reverse reaction progresses on the opposite surface having a lower $P_{O_2}$ (the $P_{O_2}$(lo) surface). Therefore, the mass transfer mechanisms through the oxides comprising the EBC can be monitored in a facile manner.

Mullite, which is often given the formula $Al_{4+2x}Si_{2-2x}O_{10-x}$, is regarded as a candidate material for multilayered EBCs because its thermal expansion coefficient is close to that of SiC and it is chemically compatible with Si-based ceramics [5]. The oxygen shielding properties of polycrystalline mullite, in which $x = 0.25$ (corresponding to $3Al_2O_3 \cdot 2SiO_2$ and referred to as so-called "stoichiometric" or stable mullite [14]) are evidently superior to those of $Yb_2Si_2O_7$ and $Yb_2SiO_5$ [10–12], although the resistance of mullite to volatilization by water vapor is far inferior to that of the Yb-silicates [15]. Oxygen permeation through these complex oxides is controlled by the inward diffusion of oxide ions and the outward diffusion of cations (*M*), such as Al ions in the case of mullite and Yb ions in the case of Yb-silicates, through grain boundaries (GBs) [10–12]. The mobility of Si ions, which comprise the majority of the complex oxides, is evidently too low to contribute to oxygen permeation. Thus, going from the SiC/SiC substrate to the top coat, an EBC typically consists of a Si-based non-oxide bond coat, a mullite layer and Yb-silicate layers. This configuration improves both the oxygen shielding properties and water vapor volatilization resistance of the overall EBC system.

Using the oxygen permeation data resulting from prior work, the oxygen permeability constants of each oxide layer constituting a multi-layered EBC, as well as the contributions of oxide ions and cations to oxygen permeation through the oxide, mass transfer parameters such as chemical potentials, GB diffusion coefficients and fluxes of both ions, can all be calculated at arbitrary temperatures and $d\mu_O$ values [8–12]. The ability to determine mass transfer through oxides provides useful structural information regarding layer arrangement, layer thickness and grain size that can assist in designing EBCs with excellent oxygen shielding characteristics together with phase stability.

When a mullite layer with superior oxygen shielding characteristics is applied to a conventional bond coat made of Si, the mullite is exposed to an extremely large $d\mu_O$ as part of the overall EBC system. The subsequent mass transfer of species diffusing through the mullite is expected to greatly affect the structural stability of the EBC. The present study assessed strategies to improve the oxygen shielding properties and phase stability of a mullite film applied to a SiC/SiC substrate through a Si-based non-oxide bond coat, using previously acquired oxygen permeation data [10] for mullite at high temperatures. The validity of these strategies was verified via experimental work at 1673 K using bilayer specimens consisting of a mullite film and a bond coat substrate, serving as model EBCs.

## 2. Design to Improve the Oxygen Shielding and Phase Stability of Mullite

In this study, the oxygen shielding obtained from a mullite film formed on a bond coat substrate was examined, employing a simplified EBC system. As described in the Introduction, the oxidation of a Si-based bond coat proceeds via the inward GB diffusion of oxide ions and the outward GB diffusion of Al ions through the exposed upper mullite film following the application of a large $d\mu_O$ at high temperatures. The transport of Al ions reduces the phase stability of the mullite in the vicinity of the junction interface between the mullite film and the underlying bond coat. Therefore, it would be beneficial to design a bond coat capable of supplying Al ions to the adjacent region in the upper mullite film that is otherwise depleted of Al.

β′-SiAlON, having the formula $Si_{6-z}Al_zO_zN_{8-z}$, is a particularly promising candidate for bond coat applications because it provides superior heat resistance compared to Si and its thermal expansion coefficient is close to that of SiC/SiC [16]. Moreover, β′-SiAlON can dissolve significant quantities of Al, and so could serve to supply Al to the mullite film. In addition, even though the concentration of Al in the β′-SiAlON would be decreased during this process, the β′-SiAlON is able to transition to

other oxynitrides during this process, such as O'-SiAlON and the X phase. This continuous variation in phase structure means that the β'-SiAlON is less likely to delay formation of $SiO_2$ and subsequent crystallization such as cristobalite [17,18]. It is well known that phase transition of cristobalite from β-type to α-type at approximately 540 K during cooling leads to significant shrinkage (3.9 vol.%) [19], resulting in initiation of microcracks at the interfaces and/or in the cristobalite layer. Furthermore, in the case of $Si_2N_2O$ that is a unit structure of O'-SiAlON having the formula $Si_{2-x}Al_xO_{1+x}N_2$, where the *x*-value varies from zero to ~0.2, the thermodynamic equilibrium $P_{O_2}$ values, in which $Si_2N_2O$ and $SiO_2$ coexist, are larger than those for Si and SiC [20,21]. Therefore, it is expected that the level of oxygen shielding required for oxide films to protect oxidation of underlying SiAlON-based ceramics can be lower than that for Si and SiC. Thus, incorporating a SiAlON bond coat is advantageous for maintaining long-term adhesion between the mullite layer and the bond coat. In the work reported herein, the effects of the bond coat material on the oxygen shielding and phase stability of the mullite film are discussed, based on comparing β'-SiAlON and Si bond coats.

When a mullite film is exposed to a $d\mu_O$ at a high temperature, such that each surface of the film is subjected to a different $P_{O_2}$, and $P_{O_2}(hi) \gg P_{O_2}(lo)$, the oxygen permeability constant, *PL*, per unit GB length can be expressed as Equation (1) [10]:

$$\frac{PL}{S_{gb}} = \frac{A_{Al}}{4S_{gb}}\left(P_{O_2}(hi)^{3/16} - P_{O_2}(lo)^{3/16}\right) + \frac{A_O}{4S_{gb}}\left(P_{O_2}(hi)^{-1/4} - P_{O_2}(lo)^{-1/4}\right) \tag{1}$$

where *P* is the oxygen permeability and *L* is the wafer thickness. The oxygen permeability term, *PL*, normalized by the GB density, $S_{gb}$, is dependent only on the GB characteristics and is unaffected by the grain size of the polycrystalline mullite. The experimental constants $A_{Al}$ and $A_O$ are related to the adsorption reactions of oxygen molecules that occur on the $P_{O_2}(hi)$ surface and subsequent movement of Al ions and oxide ions during the oxygen permeation. $A_{Al}$ and $A_O$ normalized by $S_{gb}$ are given by Equations (2) and (3), respectively [10]:

$$\frac{|A_{Al}|}{S_{gb}} = 1.139 \times 10^{-10} \exp\left(\frac{-235 \text{ kJ/mol}}{RT}\right) \tag{2}$$

and

$$\frac{|A_O|}{S_{gb}} = 3.612 \times 10^{-3} \exp\left(\frac{-518 \text{ kJ/mol}}{RT}\right) \tag{3}$$

where *R* is the gas constant, and *T* is the absolute temperature. Thus, the former and latter terms enclosed in parentheses in Equation (1) are related to the movement of Al ions and oxide ions, respectively. In addition, the $P_{O_2}(lo)$ term in Equation (1) is assumed to equal the minimum equilibrium partial pressure ($P_{O_2,eq}$) required to form $SiO_2$ via the oxidation of Si or β'-$Si_3Al_3O_3N_5$. The $P_{O_2,eq}$ values associated with this work were calculated on a thermodynamic basis using the Fact-Sage free energy minimization computer code in conjunction with various databases, including FactPS, FToxide and FTOxCN. In the case of β'-$Si_3Al_3O_3N_5$, α-$Al_2O_3$ is formed by the partial oxidation of β'-SiAlON at an equilibrium $P_{O_2}$ value below $P_{O_2,eq}$. The ongoing oxidation process eventually degrades the various oxynitride products such as O'-SiAlON and the X phase until only $SiO_2$ and mullite remain. In this study, the $P_{O_2,eq}$ required to form $SiO_2$, which would be expected to seriously damage the adhesion of the mullite film, was examined on this basis.

Figure 1 plots the $P_{O_2,eq}$ values for the oxidation of the bond coat materials Si and β'-$Si_3Al_3O_3N_5$ as functions of temperature. The line for Si in Figure 1 corresponds to the oxygen equilibrium partial pressure at the phase boundary between Si and $SiO_2$ and is in agreement with that in the Si-O volatility diagram [20]. In the case of SiAlON, the upper side corresponds to the stable region of $SiO_2$, while the lower side is equivalent to the region where any SiAlONs such as the β', O', and X phases exist. It is evident that the $P_{O_2,eq}$ values for both materials increase with increasing temperature, and that the values for the β'-$Si_3Al_3O_3N_5$ are larger than those for Si over the entire temperature range. This

indicates that applying β′-Si$_3$Al$_3$O$_3$N$_5$ can lower the oxygen shielding performance required for the upper oxide film compared to Si. The oxygen permeability constants of mullite exposed to various dμ$_O$ were calculated using Equations (1)–(3) with the $P_{O_2,eq}$ values in Figure 1.

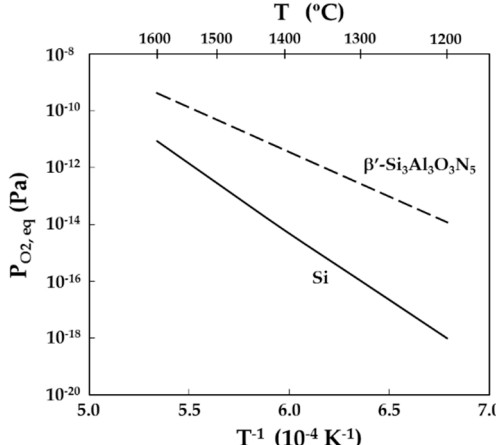

**Figure 1.** Temperature dependence of the minimum oxygen equilibrium partial pressures ($P_{O_2,eq}$) required to generate SiO$_2$ thermodynamically via the oxidation of the bond coat materials Si and SiAlON.

In the case of a mullite film on a Si bond coat (Figure 2a) with the mullite surface exposed to a $P_{O_2}$(hi) of 10$^5$ Pa, the plot showing the oxygen permeability constants related to the diffusions of both oxide ions and Al ions almost coincides with the line solely for the diffusion of oxide ions. Therefore, the oxygen permeation along the GBs in the mullite film is controlled by the inward GB diffusion of oxide ions over the entire temperature range. Using the same bilayer structure but a lower $P_{O_2}$(hi) of 1 Pa produced the data shown in Figure 2b, for which the total oxygen permeability constants and the contribution of the diffusion of oxide ions are almost equal to those in Figure 2a. However, the line associated with the diffusion of Al ions in Figure 2b drops to about one-tenth of that in Figure 2a. When the value of the $P_{O_2}$(hi) term that appears in Equation (1) was reduced to less than 10$^2$–10$^3$ times the $P_{O_2,eq}$, there was an evident decrease in the line for the diffusion of oxide ions. Thus, coating the mullite layer with a film made of an oxide such as an Yb-silicate [11,12], which provides inferior oxygen shielding, improves the phase stability of the mullite film but does not increase the overall oxygen shielding characteristics of the EBC. In the case of the mullite film on the β′-Si$_3$Al$_3$O$_3$N$_5$ bond coat with a $P_{O_2}$(hi) of 10$^5$ Pa, as shown in Figure 2c, the oxygen permeability constant is clearly decreased as a result of the significantly lower contribution of the diffusion of oxide ions. Specifically, the oxygen permeability constant due to the inward diffusion of oxide ions at 1673 K in Figure 2c was about one-fifth of the values for Figure 2a,b. It should be noted also that the contribution of the diffusion of Al ions remains much the same as in Figure 2a. The increase in the $P_{O_2,eq}$ values improves the oxygen shielding properties but not the phase stability of the mullite film.

Together, these results demonstrate that the oxygen permeation and the contribution of the diffusion of each ion through the mullite film are both significantly affected by the dμ$_O$ value. This occurs because the GB diffusion coefficients of Al ions and oxide ions change greatly as the oxygen chemical potential (μ$_O$) in the thickness direction of the mullite film varies. That is, the GB diffusion coefficients of Al ions are much larger than those of oxide ions in regions associated with high μ$_O$ values close to the $P_{O_2}$(hi) surface, while the reverse is true in regions with low μ$_O$ values close to the $P_{O_2}$(lo) surface [10]. Therefore, an EBC structure comprising a sandwich-type configuration in which the mullite film is between a SiAlON bond coat and an oxide film providing a moderate degree of oxygen shielding (such as an Yb-silicate [11,12]) is expected to be especially effective. This structure should simultaneously improve the oxygen shielding properties and phase stability of the mullite film.

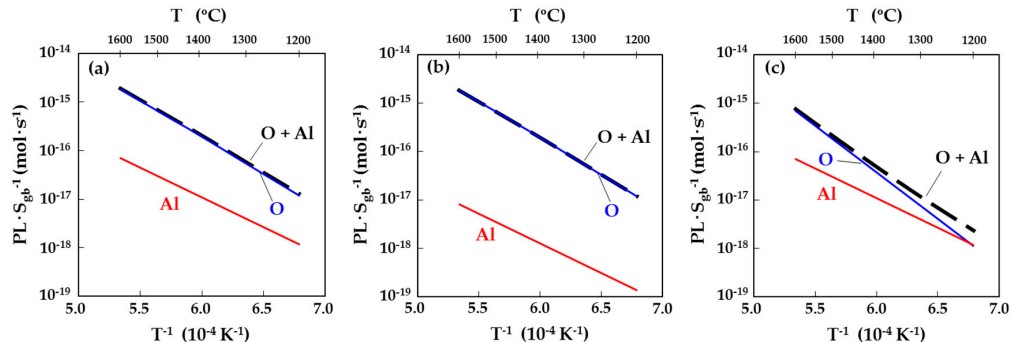

**Figure 2.** Temperature dependence of the oxygen permeability constants of mullite exposed to various oxygen potential gradients; (**a**) $P_{O_2}$(hi) = $10^5$ Pa, $P_{O_2}$(lo) = $P_{O_2,eq}$ for the oxidation of Si, (**b**) $P_{O_2}$(hi) = 1 Pa, $P_{O_2}$(lo) = $P_{O_2,eq}$ for the oxidation of Si, and (**c**) $P_{O_2}$(hi) = $10^5$ Pa, $P_{O_2}$(lo) = $P_{O_2,eq}$ for the oxidation of SiAlON.

## 3. Experimental Procedure

Figure 3 shows a schematic diagram of the procedure used to manufacture test specimens combining an oxygen shielding layer and an underlying bond layer made of Si or β′-SiAlON. In this fabrication process, raw commercially-available mullite powder (KM-101, KCM Corp., Nagoya, Japan) was molded in a uniaxial press at 20 MPa and then subjected to cold isostatic pressing at 250 MPa. The resulting green compacts were sintered in air at ambient pressure in two steps: 1573 K for 50 h followed by 2023 K for 5 h. High-resolution transmission electron microscopy (HR-TEM, TOPCON EM-002BF, Topcon Technohouse, Tokyo, Japan) indicated that there were no other phases at the GBs, meaning that there was direct contact between mullite grains. Raw β′-SiAlON powder having a *z* value of 2.94 (BSI3-001B, AG Material Inc., Taiwan) was hot pressed in a uniaxial press at 40 MPa under $N_2$ at a pressure of 0.7 MPa at 2023 K for 2 h. The relative densities of both the mullite and β′-SiAlON samples were greater than 99%. Parts with dimensions corresponding to the sections labeled types I, II and III in Figure 3, were then cut from the sintered samples and from a Si wafer (Shin-Etsu Chemical Corp. Ltd., Tokyo, Japan), and the joining surfaces of these samples were polished to a mirror-like finish. These parts were subsequently joined together using a uniaxial press at 50 MPa under reduced pressure at 1873 K for 1 h. Following this, one exposed mullite layer was ground and then polished to a final thickness of 0.25 mm. Finally, the specimens were heated in $O_2$ ($P_{O_2}$ = $10^5$ Pa) at 1673 K for 10 h. The microstructures of cross-sections of these samples in the vicinity of the bonding interfaces before and after the heat treatment were examined by scanning electron microscopy (SEM, Hitachi SU8000, Hitachi High-Technologies, Hitachinaka, Japan) and scanning transmission electron microscopy (STEM, TOPCON EM-002BF, Topcon Technohouse, Tokyo, Japan) with energy dispersive X-ray spectroscopy (EDS).

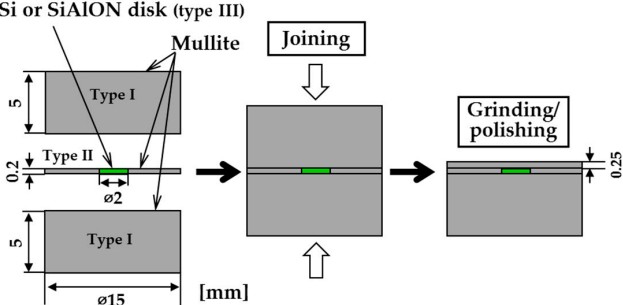

**Figure 3.** Schematic diagram of the procedure for manufacturing model samples corresponding to combinations of an oxygen shielding layer and an underlying bond layer made of Si or SiAlON.

## 4. Results and Discussion

The sample composed of a mullite film on a Si disk showed complete bonding of the mullite to the underlying Si disk in the thickness direction prior to heating, as shown in Figure 4a. No damage was observed in the vicinity of the joining interfaces in bright-field STEM images or in the corresponding EDS elemental maps of the cross-section around the interface. In contrast, after the heat treatment, damage to the mullite film near the interface was evident (Figure 4b). It is noteworthy that there was no degradation to the thicker 5 mm mullite layer at the opposite interface, which retained the same microstructure as before the heat treatment. Thus, the observed damage to the mullite is attributed to the increased driving force for the GB diffusion of Al and oxide ions resulting from the reduced film thickness. The uneven thickness of the damaged region in Figure 4b is likely related to variations in the GB density of the mullite film adjacent to the interface, because the GBs in the mullite provide paths for the rapid diffusion of ions.

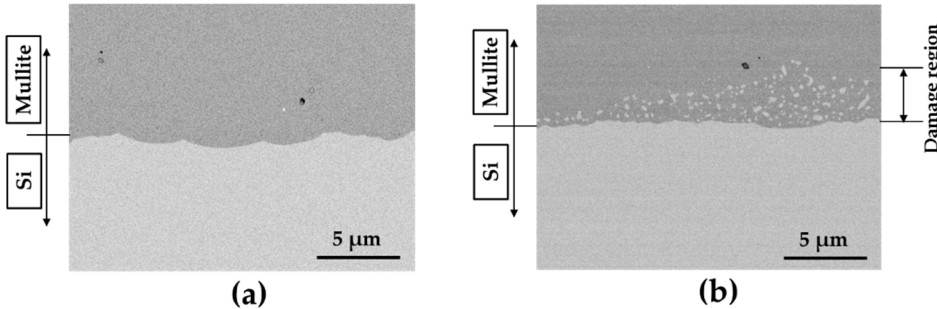

**Figure 4.** Scanning electron microscopy (SEM) images of cross-sections of the bonding interfaces of samples with a 0.25 mm thick mullite layer on an underlying Si disk (**a**) before and (**b**) after annealing in oxygen at 1673 K for 10 h.

Figure 5 presents a bright-field STEM image and the corresponding EDS elemental maps of the cross-section around the bonding interface of the annealed sample shown in Figure 4b. Because Al ions diffuse along the GBs in the mullite film in response to the application of the $d\mu_O$, both $SiO_2$-rich mullite and $SiO_2$ are expected to be formed by the decomposition of mullite near the interface. However, the damaged region shown in Figure 4b instead was found to consist of $Al_2O_3$ and Si. A Si-O phase was present only in areas enclosed by the mullite grains and located less than one micrometer from the interface. In addition, microstructural changes were not observed in the Si disk near the interface.

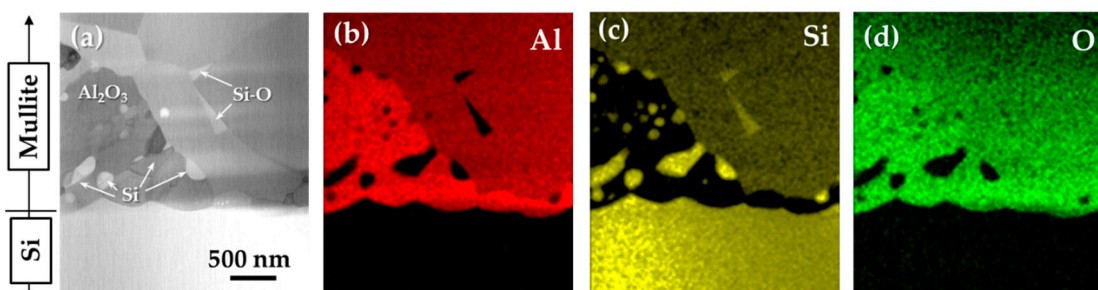

**Figure 5.** Bright-field scanning transmission electron microscopy (STEM) image and the corresponding energy dispersive X-ray spectroscopy (EDS) elemental maps of a cross-section of the bonding interface of the annealed sample shown in Figure 4b: (**a**) The bright-field image and EDS maps for (**b**) Al, (**c**) Si and (**d**) O.

Figure 6 presents a plot of the free energy curve for mullite in a $SiO_2$-$Al_2O_3$ system at 1673 K. Based on this plot, the concentration of $Al_2O_3$ in the mullite near the interface should change from an initial value, $x_i$, to a new value on the $SiO_2$-rich side of the plot, as a result of the outward GB diffusion

of Al ions. When the $Al_2O_3$ concentration reaches a critical value, $x_{cr}$, a portion of the mullite will break down to produce both $SiO_2$-rich mullite and $SiO_2$, corresponding to the Si–O phase seen in Figure 5. In this study, the specimens were fabricated by joining parts made of mullite and Si using hot pressing with a carbon heater, and so the $\mu_O$ for oxygen dissolved in the Si disk was likely extremely low. Consequently, the $SiO_2$ formed at GBs of the mullite layer in the vicinity of the bonding interface due to the outward GB diffusion of Al ions was considered to be easily reduced to produce Si particles and oxygen that immediately dissolved in the Si substrate. Thus, voids under negative pressure were thought to be formed adjacent to the Si particles derived from the mullite layer. When mullite is exposed to a total pressure of $10^{-2}$ Pa or less at 1673 K, the decomposition reaction to form $Al_2O_3$ and SiO gas proceeds thermodynamically. Therefore, the mullite located around the voids was probably decomposed, resulting in generation of the $Al_2O_3$ particles. The voids might be also filled with Si that migrated from the Si disk, since the exposure temperature was near the melting point (1687 K) of Si. On the other hand, although the amount of oxygen permeated through the mullite layer calculated from Equations (1)–(3) was much larger than the value corresponding to the solid solubility limit of oxygen in Si [22], obvious $SiO_2$ scale formation due to oxidation of the Si disk was not observed at the interface. Oxide ions diffused inward along the GBs in the mullite layer might have been captured by oxygen vacancies in the mullite layer before reaching the interface between the mullite layer and the Si disk, in which a large amount of the vacancies were created around the GBs in the mullite layer during sample preparation in the reducing atmosphere.

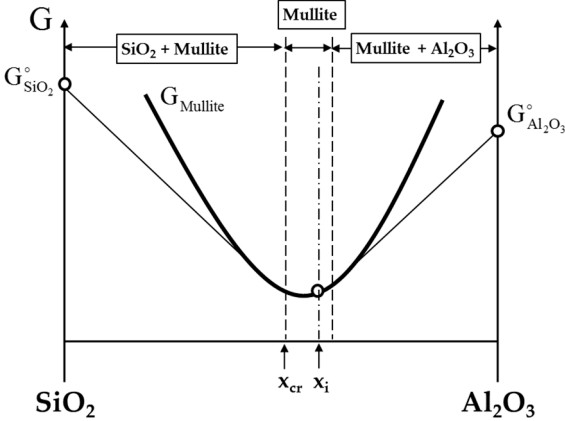

**Figure 6.** Schematic diagram of the free energy curve for mullite in a $SiO_2$-$Al_2O_3$ system at 1673 K. $x_i$ and $x_{cr}$ are the concentrations of alumina in mullite before annealing and in the mullite decomposition stage of annealing, respectively.

In the case of an actual EBC system in which a mullite layer is formed on a Si bond coat using an atmospheric plasma spray process, some ambient oxygen will dissolve in the lower Si bond coat during the high temperature coating, resulting in a sufficiently high $\mu_O$ in both the bond coat and mullite film compared with the specimens employed in this study. Therefore, a large amount of Si–O phase will be formed by decomposition of the mullite film due to the outward diffusion of Al ions and also by the oxidation of Si due to the inward diffusion of oxide ions, and will be present near the interface in the absence of oxide reduction.

Figure 7 provides TEM images of cross-sections around the bonding interface of a sample in which a 0.25 mm thick mullite layer was applied overtop of a SiAlON disk, before and after annealing in oxygen at 1673 K for 10 h. Figure 8 shows a bright-field STEM image and the corresponding EDS elemental maps of a cross-section around the bonding interface of the annealed sample shown in Figure 7b. The microstructures around the interfaces before and after the heat treatment are seen to be similar, and there is no damage to the mullite film resulting from the interdiffusion of Al and oxide ions. This result is believed to be related to the reduced inward GB diffusion of oxide ions due to the

increased $\mu_O$ value in the vicinity of the interface and to the improved phase stability of the mullite film due to the Al supply from the underlying SiAlON.

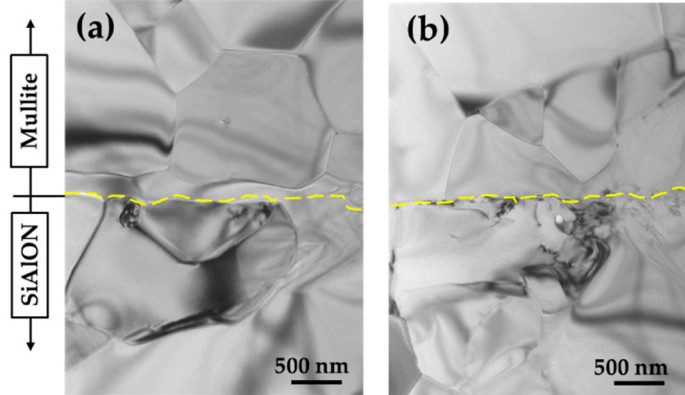

**Figure 7.** Transmission electron microscopy (TEM) images of a cross-section of the bonding interfaces of a sample with a 0.25 mm thick mullite layer on an underlying SiAlON disk (**a**) before and (**b**) after annealing in oxygen at 1673 K for 10 h.

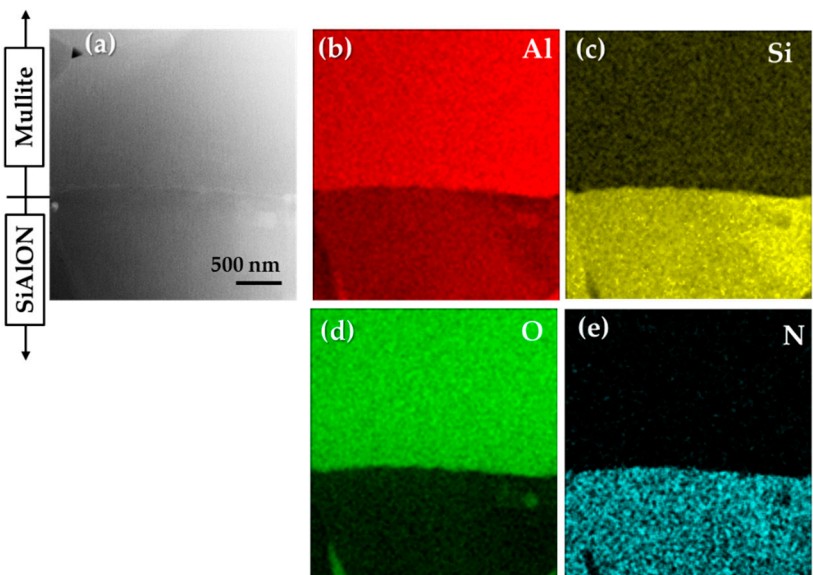

**Figure 8.** Bright-field STEM image and the corresponding EDS elemental maps of a cross-section of the bonding interface of the annealed sample shown in Figure 7b: (**a**) The bright-field image and EDS maps for (**b**) Al, (**c**) Si, (**d**) O and (**e**) N.

## 5. Conclusions

Strategies to improve the oxygen shielding properties and phase stability of mullite films on Si-based bond coats were examined and validity of these strategies was established using test models. When employing β′-SiAlON as the bond coat, the inward GB diffusion of oxide ions through the mullite film was suppressed due to an increase in the $\mu_O$ value at the interface between the mullite and the bond coat, resulting in improved oxygen shielding. Decomposition of the mullite film as a consequence of the outward diffusion of Al ions was retarded because the SiAlON served as a source of Al. Furthermore, the phase stability of the mullite was improved because the diffusion of Al through GBs was slowed due to the application of oxide films having suitable oxygen shielding characteristics.

**Author Contributions:** Conceptualization, S.K.; Methodology, S.K., N.K. and T.M.; Validation, S.K. and T.M.; Analysis, D.Y. and T.K., Investigation, S.K.; Writing—Original Draft Preparation, S.K.; Writing—Review and



Editing, S.K.; Supervision, S.K., M.T. and J.P.; Project administration, M.T. and J.P.; Funding acquisition, M.T. and J.P.

**Funding:** This work was partly supported by the Council for Science, Technology and Innovation (CSTI), the Cross-ministerial Strategic Innovation Promotion Program (SIP) "Structural Materials for Innovation" (Funding agency: JST, Japan Science and Technology Agency) and JSPS KAKENHI Grant Number JP19H05785.

**Conflicts of Interest:** The authors declare no conflict of interest.

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
