# Peer review of "Structural Stabilization of Mullite Films Exposed to Oxygen Potential Gradients at High Temperatures"

_coatings, doi:10.3390/coatings9100630_

Round 1

Reviewer 1 Report

Environmental barrier coatings are important for protection of SiC/SiC composite from degradation in high-temperature moisture environment. As successful EBCs, they must have oxygen/water vapor shielding. However, research on the oxygen permeation of EBC materials are quite few. This paper focused on research of ion diffusion and structural stabilization of bond-coat materials, which could provide scientific explanation for the mullite degradation. The paper also explored a new bond coat material, which could be used in the industry. The paper should be accepted after minor revision.

In equations (1)-(3), what do AAl and Ao represent? 1, why does PO2, eq increase with increasing temperature? PO2, eq equals minimum equilibrium partial pressure required to form SiO2 via oxidation of Si. In my understanding, the higher temperature, the lower oxygen partial pressure required to form SiO2. Page 6, the explanation on decomposition of mullite into Al2O3 and Si is still confused. The author thought the free silicon came from mullite. However, the melting point of Si is only 1410 C. The annealing temperature used in the paper was quite close to melting point of Si. At this temperature, Si became soft and readily to move. Do you think if free Si come from Si wafer?

Author Response

We wish to thank the editors and the reviewers for providing helpful comments concerning our manuscript entitled “Structural stabilization of mullite films exposed to oxygen potential gradients at high temperatures” [coatings-589971]. Please see below for our responses to the comments. Note that additions or modifications to the original manuscript are underlined in the revised manuscript to assist the editor and reviewers in evaluating our responses.

Responses to the comments of reviewer #1.

1. What do AAl and Ao represent?

The following addition has been made to the text of the revised manuscript on lines 119-121: “The experimental constants AAl and AO are related to the equilibrium constants of adsorption reactions of oxygen molecules that occur on the PO2(hi) surface during the oxygen permeation [10].”

2. Why does PO2, eq increase with increasing temperature? In my understanding, the higher temperature, the lower oxygen partial pressure required to form SiO2.

In the reaction (Si + O2 → SiO2), the thermodynamic equilibrium constant K decreases with increasing temperature. In other words, the equilibrium oxygen partial pressure increases with increasing temperature. The line for Si in Fig. 1 corresponds to the oxygen equilibrium partial pressure at the phase boundary between Si and SiO2, and is in agreement with that in the Si-O volatility diagram [20]. The reason why the SiO2 scale growth due to oxidation of Si tends to proceed from a lower oxygen partial pressure at higher temperatures is that inward diffusion of oxygen through the scale is accelerated as the temperature rises. This corresponds to a kinetic phenomenon during the passive oxidation process of Si. We discussed the phase stability of the bond coat materials thermodynamically.

The following addition has been made to the text of the revised manuscript on lines 132-135: “The line for Si in Fig. 1 corresponds to the oxygen equilibrium partial pressure at the phase boundary between Si and SiO2, and is in agreement with that in the Si-O volatility diagram [20]. In the case of SiAlON, the upper side corresponds to the stable region of SiO2, while the lower side is equivalent to the region where any SiAlONs such as the b’, O’, and X phases exist.”

The following reference was added in the revised manuscript on lines 341-342.

[20] Heuer H. A.; Lou L. K. V. Volatility diagrams for Silica, Silicon Nitride, and Silicon Carbide and their application to high-temperature decomposition and oxidation. J. Am. Ceram. Soc. 1990, 73, 2785-3218.

3. Page 6, the explanation on decomposition of mullite into Al2O3 and Si is still confused. The author thought the free silicon came from mullite. However, the melting point of Si is only 1410 ºC. The annealing temperature used in the paper was quite close to melting point of Si. At this temperature, Si became soft and readily to move. Do you think if free Si come from Si wafer?

The following addition has been made to the text of the revised manuscript on lines 237-244: “Consequently, the SiO2 formed at GBs of the mullite layer in the vicinity of the bonding interface due to the outward GB diffusion of Al ions was considered to be easily reduced to produce Si particles and oxygen that immediately dissolved in the Si substrate. Thus, voids under negative pressure were thought to be formed adjacent to the Si particles derived from the mullite layer. When mullite is exposed to a total pressure of 10-2 Pa or less at 1673 K, the decomposition reaction to form Al2O3 and SiO gas proceeds thermodynamically. Therefore, the mullite located around the voids was probably decomposed, resulting in generation of the Al2O3 particles. The voids might be also filled with Si that migrated from the Si disk, since the exposure temperature was near the melting point (1687 K) of Si.”

Reviewer 2 Report

Coatings Paper 589971 (Kitaoka, et al.) discusses modelling and experimental proof of a novel bond coat for environmental barrier coatings for SiC CMCs.  It is superb in all aspects.  It is very well written and should be published.  There are a few issues below that the authors may wish to consider.  Also, some auto-referencing errors need to be corrected. 

The oxidation behavior of SiAlON materials has been well studied.  It may be instructive to the general readers to show comparative kinetics to Si, SiC oxidation.  As I recall, SiAlON can exhibit almost no oxidation in air.  This would insure protection of SiC as a bond coat layer, as the authors have shown. 

https://scholar.google.com/scholar?start=60&q=allintitle:+sialon+oxidation&hl=en&as_sdt=0,36&as_ylo=1980&as_yhi=2000

What is the thermal expansion coefficient of SiAlON compared to SiC?

How are SiAlON coatings deposited practically?  I suspect plasma spraying would present stoichiometry problems.

Figure 2c.  While the Al flux is clearly increased for SiAlON, the important oxygen values in 2c do not appear significantly changed from a, b.  If the oxygen decrease is the important result here, it would be good to state specific values for a,b,c showing the degree of decrease at , for example, 1400 C.

Line 227.  2 sentences might help.  The term ‘decompression’ is not very clear in this context.

Line 284.  Lee K.N., van Roode M,   ? A.B. is not an author, just Lee and van Roode.

Line 317.  Kriven W.M.

Author Response

We wish to thank the editors and the reviewers for providing helpful comments concerning our manuscript entitled “Structural stabilization of mullite films exposed to oxygen potential gradients at high temperatures” [coatings-589971]. Please see below for our responses to the comments. Note that additions or modifications to the original manuscript are underlined in the revised manuscript to assist the editor and reviewers in evaluating our responses.

Responses to the comments of reviewer #2.

1. The oxidation behavior of SiAlON materials has been well studied.  It may be instructive to the general readers to show comparative kinetics to Si, SiC oxidation.  As I recall, SiAlON can exhibit almost no oxidation in air.  This would insure protection of SiC as a bond coat layer, as the authors have shown. 

https://scholar.google.com/scholar?start=60&q=allintitle:+sialon+oxidation&hl=en&as_sdt=0,36&as_ylo=1980&as_yhi=2000

The oxidation rate of Si3N4 ceramics, including SiAlON, strongly depends on their sintering aids. Thus, it is difficult to directly compare the oxidation behavior of b'-SiAlON (z = 3), which was the subject of this study, with those for Si and SiC. Therefore, the following addition based on a thermodynamic viewpoint has been made to the text of the revised manuscript on lines 99-104: “Furthermore, in the case of Si2N2O that is a unit structure of O’-SiAlON having the formula Si2-xAlxO1+xN2, where the x-value varies from zero to ~0.2, the thermodynamic equilibrium PO2 values, in which Si2N2O and SiO2 coexist, are larger than those for Si and SiC [20, 21]. Therefore, it is expected that the level of oxygen shielding required for oxide films to protect oxidation of underlying SiAlON-based ceramics can be lower than that for Si and SiC”.

The following references were added in the revised manuscript on lines 341-344.

[20] Heuer H. A.; Lou L. K. V. Volatility diagrams for Silica, Silicon Nitride, and Silicon Carbide and their application to high-temperature decomposition and oxidation. J. Am. Ceram. Soc. 1990, 73, 2785-3218.

[21] Jacobson S. N. Corrosion of silicon-based ceramics in combustion environments. J. Am. Ceram. Soc. 1993, 76, 3-28.

2. What is the thermal expansion coefficient of SiAlON compared to SiC?

The thermal expansion coefficients (RT-1673K) of b'-SiAlON (z = 3) and SiC are 4.0×10-6 K-1 and 5.0×10-6 K-1, respectively. The thermal expansion coefficient of SiAlON increases slightly with increasing z [16].

3. How are SiAlON coatings deposited practically? I suspect plasma spraying would present stoichiometry problems.

A SiAlON layer has been successfully formed on a SiC/SiC substrate by dual electron beam–physical vapor deposition (EB-PVD), in which the coated surface was heated at a temperature above 1273 K in a trace amount of NH3 gas by irradiation of a direct diode laser (wavelength of 915 nm) to promote full crystallization and densification. In this technique, two targets such as Si and Al2O3 located in the deposition chamber are evaporated using two electron guns regulated individually, which enables independent control of the vapor pressures of the gases generated from the targets. For the adhesion of a multilayered EBC, including the SiAlON bond layer and the upper mullite oxygen shielding layer formed based on the design guideline of EBC, please refer to the paper in the special issue of EBCs, which was recently submitted by Kawai E. et al (coatings-604600).

4. Figure 2c.  While the Al flux is clearly increased for SiAlON, the important oxygen values in 2c do not appear significantly changed from a, b.  If the oxygen decrease is the important result here, it would be good to state specific values for a, b, c showing the degree of decrease at , for example, 1400 ºC.

In accordance with the reviewer’s comment, the following sentence has been changed in the revised manuscript on lines 151-152: “the line associated with the diffusion of Al ions in Fig. 2(b) drops to about one-tenth of that in Fig. 2(a).” Also, the following addition has been made to the text of the revised manuscript on p.4, lines 159 - 161: “Specifically, the oxygen permeability constant due to the inward diffusion of oxide ions at 1673 K in Fig. 2(c) was about one-fifth of the values for Figs. 2(a) and (b).”

5. Line 227.  2 sentences might help.  The term ‘decompression’ is not very clear in this context.

In accordance with the reviewer’s comment, the following sentences were changed in the revised manuscript on lines 237-251: “Consequently, the SiO2 formed at GBs of the mullite layer in the vicinity of the bonding interface due to the outward GB diffusion of Al ions was considered to be easily reduced to produce Si particles and oxygen that immediately dissolved in the Si substrate. Thus, voids under negative pressure were thought to be formed adjacent to the Si particles derived from the mullite layer. When mullite is exposed to a total pressure of 10-2 Pa or less at 1673 K, the decomposition reaction to form Al2O3 and SiO gas proceeds thermodynamically. Therefore, the mullite located around the voids was probably decomposed, resulting in generation of the Al2O3 particles. The voids might be also filled with Si that migrated from the Si disk, since the exposure temperature was near the melting point (1687 K) of Si. On the other hand, although the amount of oxygen permeated through the mullite layer calculated from Eqs. (1)-(3) was much larger than the value corresponding to the solid solubility limit of oxygen in Si [22], obvious SiO2 scale formation due to oxidation of the Si disk was not observed at the interface. Oxide ions diffused inward along the GBs in the mullite layer might have been captured by oxygen vacancies in the mullite layer before reaching the interface between the mullite layer and the Si disk, in which a large amount of the vacancies were created around the GBs in the mullite layer during sample preparation in the reducing atmosphere”.

6. Line 294.  Lee K.N., van Roode M,   ? A.B. is not an author, just Lee and van Roode.

Line 327.  Kriven W.M.

In accordance with the reviewer’s comment, the author names were corrected in the revised manuscript on lines 295 and 328.